# Investigation of an In-Line Slot Waveguide Sensor Built in a Tapered D-Shaped Silicon-Cored Fiber

**DOI:** 10.3390/s21237832

**Published:** 2021-11-25

**Authors:** Kai-Ju Lin, Lon A. Wang

**Affiliations:** Graduate Institute of Photonics and Optoelectronics, National Taiwan University, Taipei 106, Taiwan; r07941083@ntu.edu.tw

**Keywords:** silicon cored fibers, D-shaped, slot waveguides, sensors

## Abstract

An in-line slot waveguide sensor built in a polished flat platform of a D-shaped silicon cored fiber with a taper coupled region is proposed and investigated thoroughly. Simulation results show that the single-mode light field sustained in the silicon cored fiber can be efficiently transferred to the slot waveguides through the tapered region. The geometry parameters of the slot waveguide sensors are optimized to have the corresponding highest power confinement factors and the resultant sensor sensitivities. The three-slot waveguide sensor is found to have the best performance among one-, two- and three-slot waveguides at the mid-IR wavelength.

## 1. Introduction

Rib and strip silicon waveguides used as biosensors are commonly constructed in the form of a Fabry–Perot cavity [1], a ring-cavity [2], or a Bragg grating [2]. In these structures, most light fields are strongly confined in silicon core due to high refractive index contrast between core and cladding, leading to weak light interaction with the analyte in the surrounding environment. To overcome this limit, Almeida et al. first proposed the slot waveguide to confine and enhance the light field in a submicron slot which consisted of a low refractive index material [3]. The light–analyte interaction was enhanced in the slot region compared to the rib/strip silicon waveguides. Since then, slot waveguides have been developed with significant increases in sensing sensitivity for many applications such as label-free molecule detection [4,5].

The key advantage of optical fiber technologies is the ability to connect components in a seamless and robust structure. Although the application of silica fibers has been studied thoroughly, the desire to extend the working wavelength into the mid-infrared region is motivating the search for new materials for this platform. Silicon cored fibers (SCFs) have drawn lots of attention, and many works have been reported because of their unique characteristics, such as large optical nonlinearity [6,7], and electro-photonics [8,9]. Unlike silica fibers, the silicon core of an SCF does not possess large photosensitivity for making a short-period or a long-period fiber grating, limiting its usage for sensing applications. Only a few SCF-based sensors have been reported [10,11], but none to the best of our knowledge for biosensors. To that end, we propose to side-polish an SCF to form a D-shaped SCF (DSCF) and use its flat surface region as a platform for building biosensors. The DSCF has been experimentally demonstrated to build a waveguide photodetector [9], proving the flat surface could be controlled smooth enough for opto-electronic integration. Here we show the design and the simulation results of slot waveguide sensors built in a tapered DSCF to demonstrate the feasibility of building an in-line SCF biosensor, which cannot be fabricated on the conventional mid-infrared fibers or silica fibers due to the refractive index difference between core and the air is smaller.

The description of this work is arranged as follows. First, we propose a different coupling scheme instead of the grating coupling commonly seen in silicon photonics. We then show that the single-mode operation can be maintained when propagating through the tapered and the D-shaped regions. Secondly, the key parameters such as slot width, slot separation, and the number of slots for the best performance of slot waveguides are optimized. Thirdly, we show the feasibility of using an in-line fiber slot waveguide as an index sensor intended for biomedical applications.

## 2. Coupling Method

A slot waveguide usually has slot width in the order of a few tens of nanometers. It is embedded in a silicon waveguide structure on a silicon photonic platform, usually in a silicon-on-insulator configuration. To efficiently couple light into such a small slot is a challenging issue. Generally, two couplings are involved: light presumably from an external optical glass fiber needs to be first coupled to a silicon waveguide structure. Then the light goes through a second coupling into a slot waveguide. As a common practice in silicon photonics, a grating is adopted for the first light coupling as the input port of the silicon waveguide [12]. In such a coupling scheme, an input fiber must be placed vertically with the slanted angle and phase-matched to the waveguide plane to achieve a high coupling efficiency. As pointed out by Reference [13], such a vertical or non-coplanar structure may increase packaging complexity compared to the parallel or coplanar structure usually found in the light coupling between an edge-emitting semiconductor laser and an optical fiber. For the second coupling between silicon waveguide and slot waveguide, several exemplary schemes have been reported, such as tapered strip waveguide coupler [12] and strip-slot coupler [14].

Figure 1 shows our light coupling scheme using a tapered DSCF. Instead of using a grating coupling scheme, a commercial mid-IR SMF is directly coupled to a tapered DSCF. A tapered DSCF is composed of a tapered SCF with its D-shaped flat surface located in the waist region. Assume the light wavelength is 3 µm, where many biomarkers have their fingerprints [15]. The silicon core diameter of our SCF made by the molten core drawing method was usually larger than that required for the single-mode condition, resulting in multiple modes propagating in the core at this wavelength. Therefore, the single-mode light field from a mid-IR SMF must be sustained along the tapered DSCF and then be coupled to a slot waveguide situated in the flat surface of DSCF.

Since the last decade, there has been a growing interest in avoiding exciting higher-order modes when coupling a Gaussian beam into a multimode fiber for increasing modal bandwidth and transmission speed, the so-called mode division multiplexing technique. These coupling methods include off-set launch technique [16], selective mode excitation techniques, and mode field matched center launching [17], which in turn also provide ways for sustaining single-mode light propagation in the SCF. For example, it has been reported that the preset diameter of SCF could be determined by using the method of mode field matched center launching so that the launched light can maintain single-mode propagation throughout the SCF [18]. Similarly, the optimal silicon core size which had the most overlapping with the fundamental mode of a mid-IR SMF could be obtained using the software Lumerical Mode Solution. The result showed that when the SMF had a core diameter of 9 μm, the optimal diameter of the connecting SCF was 14 μm so that 80% of the incident field could be coupled into the fundamental mode of the SCF. To reduce the Fresnel reflection when the SCF and the mid-IR SMF are connected, a micron-structured surface on the end of SCF may suffice with low coupling losses [19,20].

For the second light coupling, a tapered SCF is polished at a certain waist location as shown in Figure 1 but magnified in Figure 2 to create a DSCF in which slot waveguides are built in the flat silicon region to serve as an in-line fiber-based biosensor. An SCF can be tapered to the desired diameter by using, for example, an oxyhydrogen flame system. The tapered SCF is then fixed on a V-groove whose depth is designed to achieve the polished depth of a DSCF. The flat platform of the DSCF is generated by using a grinding wheel system for precise polish depth control. Afterward, the narrow slot can be fabricated in DSCF by using electron beam or helium beam lithography for patterning and followed by reactive ion etching.

To confirm that the incident light can maintain single-mode throughout a tapered DSCF, we used the beam propagation method to simulate the evolution of the light field inside the tapered DSCF as shown in Figure 2. The original 1.5 µm core diameter was polished to 0.75 µm thickness to form a flat surface and an ensued tapering region. Figure 3a shows the evolution of the light field of yz-plane from the incident facet to the distance of 100/200/350 µm across a tapering length of 200 µm and a tapering angle of about 0.2 degrees. The light field is shown gradually compressed towards the polished surface and results in light enhancement. Note that in Figure 3b–e the light field remains single-mode throughout the tapering and flat regions, facilitating the design of an embedded slot waveguide (SW) sensor in the flat area.

## 3. Simulation Results and Discussion

The principle of generating slot mode in the low index region is due to the significant discontinuity of the electric field of the high refractive index contrast in the interface. The electric field is proportional to the ratio of $n_{H}^{2}/n_{s}^{2}$ [3], where *n_H_* and *n_s_* are the refractive indices of core material and slot region, respectively. The TE-mode light in the slot region is thus enhanced, making it possible to realize a slot waveguide in a DSCF.

Rsoft Beamprog. was used to confirm that the guided core mode in the DSCF could be coupled into slot mode of SW in DSCF. The simulation model of a slot waveguide built in a DSCF and the cross-section of the slot region are shown in Figure 4a,b, respectively. The core diameter was 1.5 µm and the depth of SW was around 0.75 µm. Figure 5a shows that the electric field distribution of incident light propagating through the SW in the DSCF. Note that light was indeed coupled into the slot waveguide with 140 nm width inside the DSCF as shown in Figure 5b in which the field distribution was recorded by monitoring M2 along xy-plane. Figure 6 shows the evolution of norm electric field. The red, blue and green curves were the monitored |E| values at locations M1, M2 and M3, respectively. It is clearly seen that the initially guided mode ($\left|E_{H}\right|)$ increased significantly when coupled into the slot mode, from 0.4 to 4.7 at steady state, indicating the electric field enhancement in the SW region. The result was consistent with the theory that the norm electric field amplitude of slot mode could be 11 times larger than that of the guided mode because $\left|\frac{E_{S}}{E_{H}}\right|=n_{H}^{2}/n_{s}^{2}$ ($n_{H}\sim 3.4,n_{s}\sim 1$), where $\left|E_{s}\right|$ is the norm electric field in the SW region and $\left|E_{H}\right|$. is the norm electric field in the silicon core.

Ideally, the polished loss of DSCF can be reduced to 0.1 dB if the tapering length is long enough when surface roughness and material absorption are neglected. The result of DSCF’s transmitted intensity when polished from 1.5 µm to 0.75 µm with different tapering lengths is shown in Figure 7. Regardless of the profile of the transition region, as long as the tapering length (*z*-axis) was long enough, the transmission loss could be reduced to 0.1 dB and the excitation of higher-order modes can be avoided.

Furthermore, we used 2D FDTD to study the SW in the DSCF as shown in Figure 4b. For sensing applications, the working principle of a slot waveguide is based on the variation of absorption. According to Beer-Lambert’s law [21], the output intensity, *I*, can be expressed as
(1)$$I=I_{0}e^{-\Gamma \varepsilon Cl-\alpha l}$$
where *I*_0_ is the input intensity, *ε* the absorption coefficient of the analyte, *C* the concentration of the analyte, α the intrinsic loss of DSCF, *l* the length of sensing area, and *Γ* power confinement factor (PCF). *Γ* is defined as [21]
(2)$$\Gamma =\frac{\iint_{s}P_{z}dxdy}{\iint_{t}P_{z}dxdy}$$
where *Pz* is the *z* component of the Poynting vector normal to the DSCF cross section, *s* for slot region, and *t* for total area of device. The absorption sensitivity of the sensor (S) can be calculated by differentiating normalized intensity (*I_norm_ = I/I*_0_) with respect to C [21]
(3)$$S=\frac{d\left(I∕I_{0}\right)}{dC}=-\Gamma \varepsilon Cle^{\left(-\Gamma \varepsilon Cl-\alpha l\right)}=-\Gamma \varepsilon lI_{norm}$$

Thus the sensitivity of SW in DSCF strongly depends on PCF which is then maximized with respect to different slot widths by fixing other parameters. Figure 8 shows the high expression of norm electric field distribution with 120 nm slot width by assuming TE polarization at the operating wavelength of 3 µm. It is clearly seen that the electric field inside the slot region was extremely higher than the core guided mode. The result of PCF of one SW in the DSCF is shown in Figure 9a PCF gradually increased with increasing slot width from 60 nm, and reached the maximum PCF of 49.1% at 120 nm. When the slot width was larger than penetration length (~100 nm), the electric field became nonuniform in slot region and PCF started to decrease. Note that the normalized power (NP) ($\frac{\iint_{s}P_{z}dxdy}{Area_{slot}}\left(\frac{W}{m^{2}}\right))\;$ in the slot region slightly decreased with increasing slot width. The result was comparable to what was reported in the channel slot waveguide whose PCF was around 49% at 1550 nm [5].

In addition, we study the effect of the number of slots. The 2D FDTD was used again to calculate the optical properties of the SW in DSCF with two/three slots while the dimension of the SW in DSCF remained the same as shown in Figure 4. The slot width of SW in DSCF was set at the optimal value 120 nm as mentioned previously. We only changed the separation between slots to optimize PCF, and the result is shown in Figure 9b. The highest PCF occurred at the separation equal to 250 nm whose PCF was 65% in the two-slot case and 81% in the three-slot case. Figure 10a,c show the power distribution of the two/three SWs in DSCF when the separation was optimized. The reason for poor PCF in Figure 10b was due to the larger separation than the size of the guided fundamental mode, resulting in poor overlapping between DSCF’s guided mode and the slot mode so that most guided field could not couple into the slot region. Similarly, if the guided mode is of higher order, PCF would be much lower due to poor overlapping of the fields. It is also noted that as the separation exceeded the peak, PCF decreased dramatically.

Usually the sensitivity of waveguide can be defined as absorption loss [21], variation of effective index [22], or the wavelength shift [23]. The variation of effective index can be manifested in the change of transmission power or the shift of wavelength. Therefore, the sensitivity of the SW in a DSCF (*S_sw_*) can be defined as follows [22]:
(4)$$\Delta n=\frac{n_{b}-n_{b,ref}}{n_{b,ref}}$$
(5)$$\Delta n_{eff}=\frac{n_{eff}-n_{eff,ref}}{n_{eff,ref}}$$
(6)$$S_{sw}=\frac{\Delta n_{eff}}{\Delta n}$$
where *n_b_* is the refractive index of analyte to be characterized; *n_b,ref_* the reference refractive index. In the calculation of $S_{sw},$ we set water as reference (*n_b,ref_* = 1.37 at 3 μm); *n_eff_* the effective index of the slot waveguide; *n_eff,ref_* is the *n_eff_* in water. Figure 11 shows the variation of $S_{sw}$ and PCF when *n_b_* varies from 1.38 to 1.51. The solid lines correspond to slot waveguide sensitivity and the dotted lines correspond to the PCF. The black lines are the cases of one-slot, the orange two-slot, and the blue three-slot waveguides. It is seen that $S_{sw}$ was significantly enhanced with increasing number of slots. The trend of $S_{sw}$ variation is similar to the variation of PCF, indicating that the sensitivity is significantly affected by PCF. For comparison, the sensitivity (*S_sw_*) for conventional slot waveguide of slot width = 100 nm, waveguide width = 200 nm and waveguide height = 220 nm is shown in Figure 12 with the red curve. The ridge slot waveguide has the same dimension as the conventional slot waveguide with an unetched height of 20 nm is shown in the blue curve [24]. The results are compared with the optimized three-slot waveguide in the tapered DSCF with the optimal slot width of 120 nm and separation of 200 nm, as shown in the black curve in Figure 12. It is seen that the proposed SW in a tapered DSCF is more sensitive than the ridge slot waveguide but lower than the conventional slot waveguide. The advantage of our proposed slot waveguide is that instead of the channel or ridge geometry, the fiber geometry provides a more convenient coupling method, for instance, free-space coupling or splicing with an optical fiber. Though more slots would lead to higher sensitivities, the incurred cost may not be worthy from practical viewpoints. For example, adding more slots would naturally increase fabrication complexity, weaken the structure’s mechanical strength, and increase optical loss. Therefore, we limit the number of SW no more than three.

## 4. Conclusions

The proposed method of light coupling was capable of sustaining a single-mode light field from a mid-IR SMF to the slot waveguide in a tapered DSCF. The field enhancement in the slot waveguide was confirmed, and PCFs, the critical parameter affecting the sensitivity of SW sensor, were optimized with slot widths by using both 2D and 3D FDTD for one-, two- and three-slot waveguides. The PCF of the one-slot waveguide in the tapered DSCF would reach 49.1%, comparable to the traditional channel slot waveguide. The highest PCF (81.5%) was found for the three-slot waveguide in the tapered DSCF with the optimal slot width of 120 nm and separation of 200 nm. For the application of detecting the refractive index change of the analyte, the sensitivity of the slot waveguides in the tapered DSCF was found to be strongly related to PCF and increased significantly with slot numbers. Three-slot waveguide in the tapered DSCF was therefore considered the best candidate for realization. Our proposed SW in a DSCF, being in a fiber geometry, has the potential to be directly accessed with the existing mid-infrared optical fiber, paving the way for an all-fiber system with potentially much greater functionality.

## Figures and Tables

**Figure 1 sensors-21-07832-f001:**
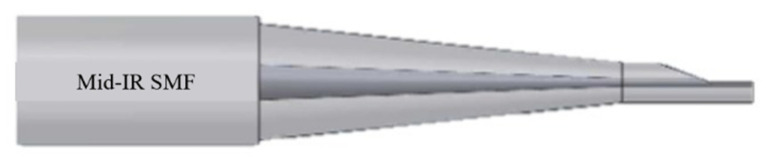
The scheme of light coupling from a mid-IR SMF to a DSCF through a tapered SCF region.

**Figure 2 sensors-21-07832-f002:**
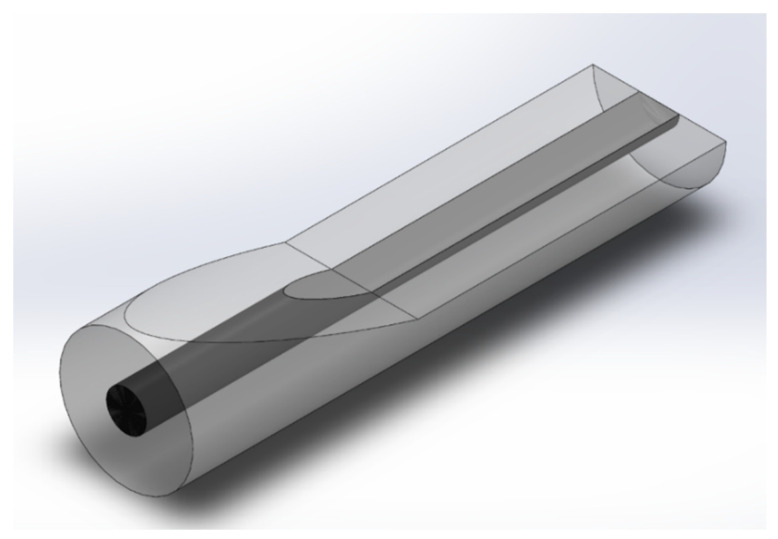
Model of our proposed DSCF.

**Figure 3 sensors-21-07832-f003:**
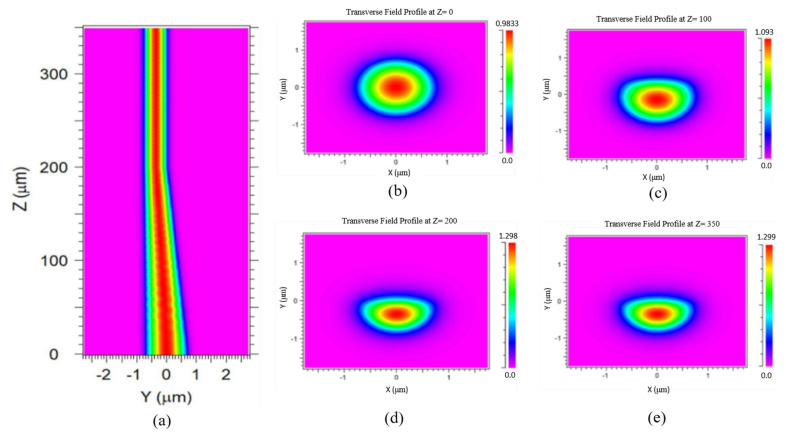
(**a**) The evolution of light field propagating through the DSCF *y–z*-plane, and light field of *x–y*-plane at (**b**) *z* = 0 µm, (**c**) 100 µm, (**d**) 200 µm, and (**e**) 350 µm.

**Figure 4 sensors-21-07832-f004:**
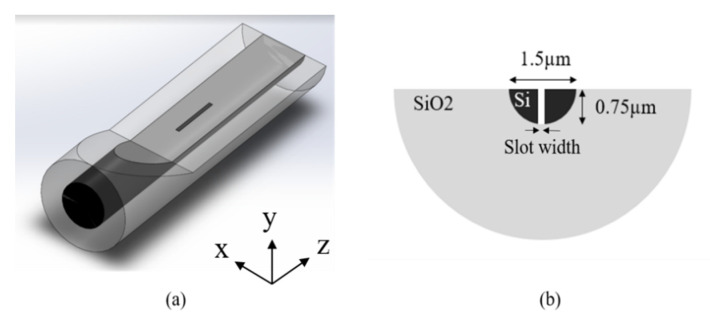
(**a**) The simulation model of an SW in DSCF. (**b**) The cross section of the slot region.

**Figure 5 sensors-21-07832-f005:**
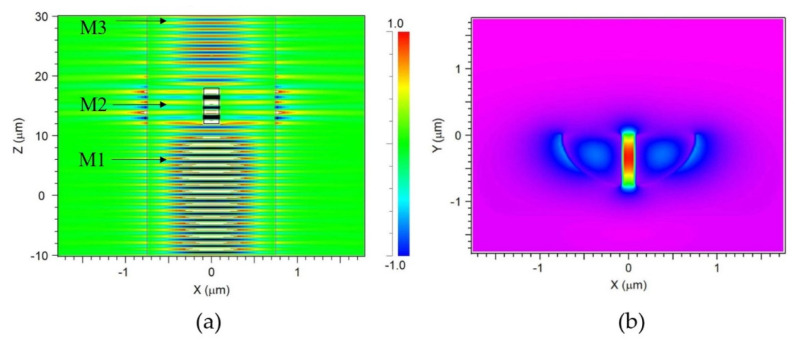
(**a**) The electric field distribution of incident light propagating through the SW in DSCF whose silicon core diameter was 1.5 μm. (**b**) The electric field of the cross section of the slot region.

**Figure 6 sensors-21-07832-f006:**
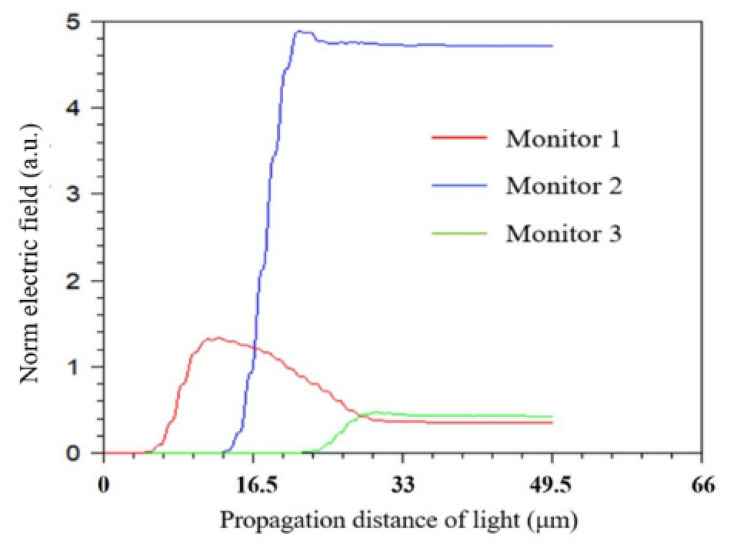
The evolution of norm electric field of SW in DSCF. The red, blue and green curves were the monitored |E| value at location M1, M2 and M3, respectively.

**Figure 7 sensors-21-07832-f007:**
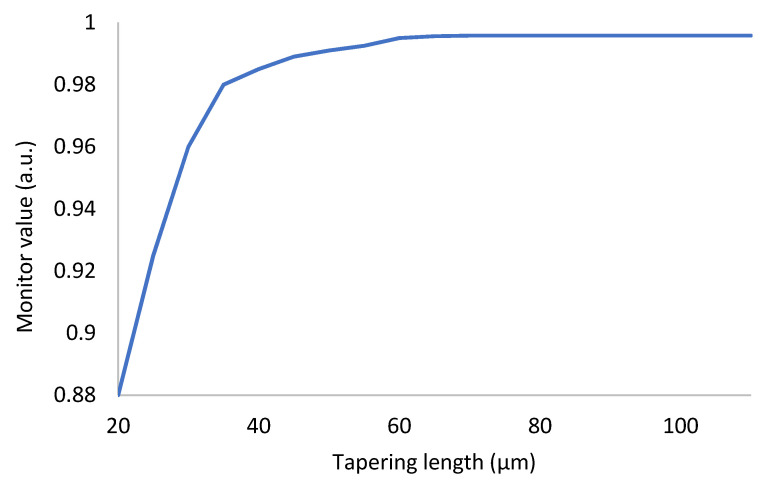
The results of DSCF’s transmitted light when polished from 1.5 µm to 0.75 µm with different tapering lengths.

**Figure 8 sensors-21-07832-f008:**
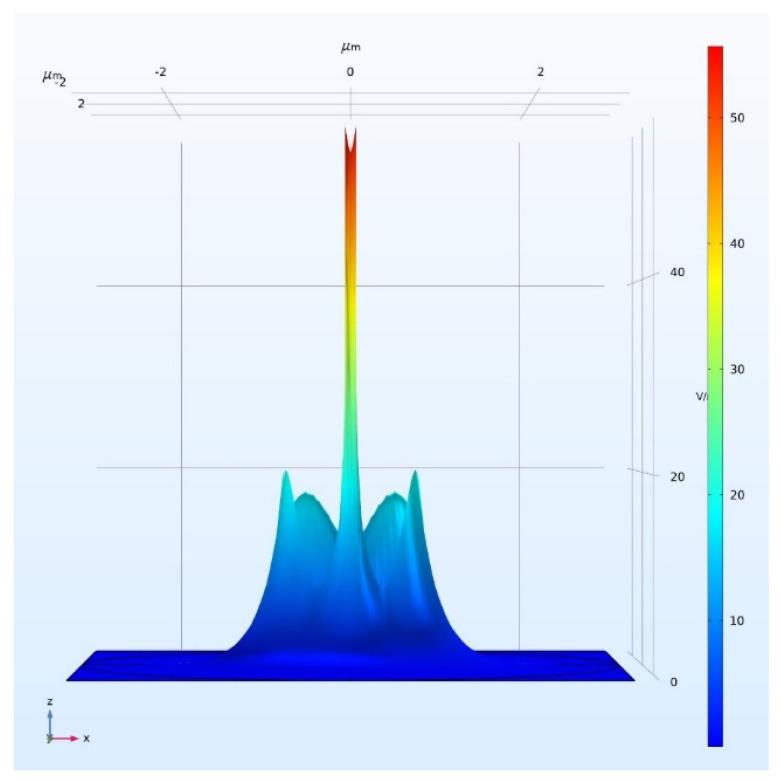
The high expression of norm electric field distribution with 120 nm slot width by assuming TE polarization at 3 µm wavelength.

**Figure 9 sensors-21-07832-f009:**
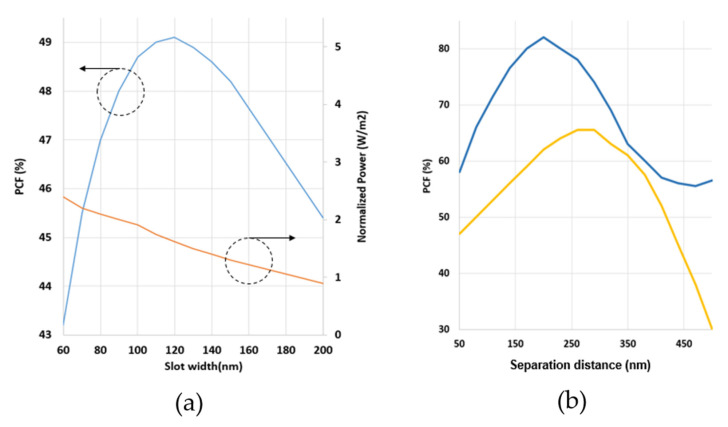
(**a**) Simulation result of one SW in DSCF. The blue curve PCF and orange curve NP vary with slot widths. (**b**) PCF of the two (yellow curve) and three (blue curve) SW in DSCF with varying separation distance.

**Figure 10 sensors-21-07832-f010:**
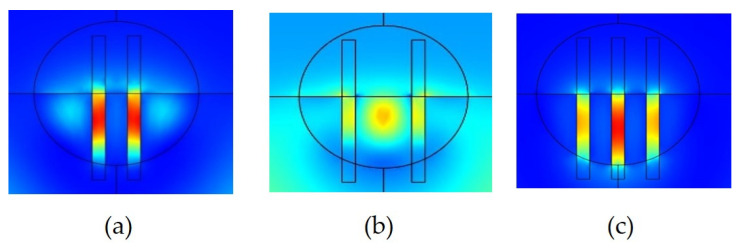
The power distribution of two SWs in DSCF when separation was (**a**) 250 nm and (**b**) 500 nm. (**c**) The power distribution of three SWs in DSCF when the PCF was the highest.

**Figure 11 sensors-21-07832-f011:**
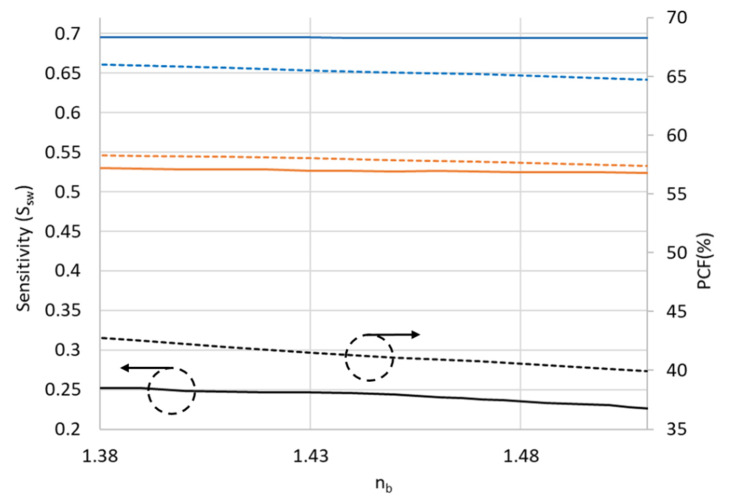
The variation of *S_sw_* when *n_b_* change from 1.34 to 1.51. The solid lines correspond to sensitivity and the dotted lines correspond to the power confinement factor. One, two and three slots are represented by black, orange and blue colors, respectively.

**Figure 12 sensors-21-07832-f012:**
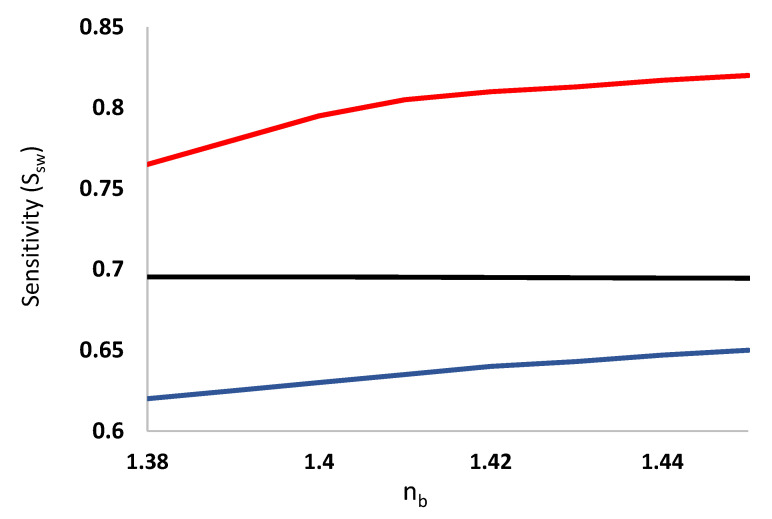
The sensitivity of the slot waveguides versus the refractive index of analyte. The red curve indicates the conventional slot waveguide with 200 nm waveguide width, 100 nm slot width and 220 nm waveguide height. The blue curve indicates the ridge slot wavguide which has the same dimension as conventional slot waveguide with an unetched height of 20 nm. The black curve indicates our proposed SW in a tapered DSCF.

## Data Availability

Not applicable.

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
