# Peer review of "Investigation of an In-Line Slot Waveguide Sensor Built in a Tapered D-Shaped Silicon-Cored Fiber"

_sensors, 2021, doi:10.3390/s21237832_

Round 1

Reviewer 1 Report

In this submission entitled "Investigation of an In-Line Slot Waveguide Sensor Built in a Tapered D-Shaped Silicon-Cored Fiber", the authors proposed an in-line slot waveguide sensor built in a polished flat platform of a D-shaped silicon cored fiber with a taper coupled region. It is found that the single-mode light field sustained in the silicon cored fiber can be efficiently transferred to the slot waveguides through the tapered region. The sensing performances of the proposed structure were also investigated.

This reviewer has the following comments and suggestions:

  1. A comparison between the sensitivities of the proposed structures and other reported sensors should be made in the manuscript.

  1. Some important conclusions or opinions in the article do not have authoritative papers for reference. Such as in Section 3, “The electric field is proportional to the ratio of ?H2 /?s2, …”.

  1. The effect of the length of the tapered region on the efficiency of the light field transfer and the sensitivity should be discussed.

  1. Can we get higher efficiency and sensitivity with more slots, for example, four and five slots?

Author Response

Dear reviewer

We are very grateful to receive your precious review comments. Please see the attachment of our answers to your questions.

Reviewer 2 Report

The manuscript proposed a tapered D-shaped fiber for biomedical sensing in terms of the refractive index. The theoretical analysis of the proposed structure was verified by simulations. Single-mode operation can be maintained after the light transmits the taper and D-shaped region. However, some of the contents need to be clarified. I recommend it to be published on Sensors after minor revisions. See below.

  1. The advantages of the proposed structure were not clearly mentioned either in the introduction or the conclusion/discussion. The only parameter mentioned was a PCF comparable to the conventional structures.
  2. The structure was designed for sensing refractive index change in biomedical applications. However, the significance of this sensing was not mentioned. It needs to be added to the introduction/discussion.

Author Response

(The authors gave the same response as above.)

Round 2

Reviewer 1 Report

The authors revised their manuscript, according to the referee's suggestions.